# Stemming the Rise of Antibiotic Use for Community-Acquired Acute Respiratory Infections during COVID-19 Pandemic

**DOI:** 10.3390/antibiotics11070846

**Published:** 2022-06-24

**Authors:** Shena Y. C. Lim, Yvonne P. Zhou, Daphne Yii, De Zhi Chin, Kai Chee Hung, Lai Wei Lee, Jia Le Lim, Li Wen Loo, Narendran Koomanan, Nathalie Grace Chua, Yixin Liew, Benjamin P. Z. Cherng, Siew Yee Thien, Winnie H. L. Lee, Andrea L. H. Kwa, Shimin J. Chung

**Affiliations:** 1Department of Pharmacy, Singapore General Hospital, Singapore 169608, Singapore; shena.lim.y.c@sgh.com.sg (S.Y.C.L.); yvonne.zhou.p.j@sgh.com.sg (Y.P.Z.); daphne.yii.y.c@sgh.com.sg (D.Y.); hung.kai.chee@sgh.com.sg (K.C.H.); lee.lai.wei@sgh.com.sg (L.W.L.); lim.jia.le@sgh.com.sg (J.L.L.); loo.li.wen@sgh.com.sg (L.W.L.); narendran.koomanan@sgh.com.sg (N.K.); nathalie.grace.sy.chua@sgh.com.sg (N.G.C.); lie.yi.xin@sgh.com.sg (Y.L.); winnie.lee.h.l@sgh.com.sg (W.H.L.L.); andrea.kwa.l.h@sgh.com.sg (A.L.H.K.); 2Department of Clinical Quality and Performance Management, Singapore General Hospital, Singapore 169608, Singapore; chin.de.zhi@sgh.com.sg; 3Department of Infectious Diseases, Singapore General Hospital, Singapore 169608, Singapore; benjamin.cherng.p.z@singhealth.com.sg (B.P.Z.C.); thien.siew.yee@singhealth.com.sg (S.Y.T.)

**Keywords:** acute respiratory tract infection, antimicrobial stewardship, COVID-19, antimicrobial resistance

## Abstract

At the start of the COVID-19 pandemic, there was an increase in the use of antibiotics for the treatment of community-acquired respiratory tract infection (CA-ARI) in patients admitted for suspected or confirmed COVID-19, raising concerns for misuse. These antibiotics are not under the usual purview of the antimicrobial stewardship unit (ASU). Serum procalcitonin, a biomarker to distinguish viral from bacterial infections, can be used to guide antibiotic recommendations in suspected lower respiratory tract infection. We modified our stewardship approach, and used a procalcitonin-guided strategy to identify “high yield” interventions for audits in patients admitted with CA-ARI. With this approach, there was an increase in the proportion of patients with antibiotics discontinued within 4 days (16.5% vs. 34.9%, *p* < 0.001), and the overall duration of antibiotic therapy was significantly shorter [7 (6–8) vs. 6 (3–8) days, *p* < 0.001]. There was a significant decrease in patients with intravenous-to-oral switch of antibiotics to “complete the course” (45.3% vs. 34.4%, *p* < 0.05). Of the patients who had antibiotics discontinued, none were restarted on antibiotics within 48 h, and there was no-30-day readmission or 30-day mortality attributed to respiratory infection. This study illustrates the importance of the antimicrobial stewardship during the pandemic and the need for ASU to remain attuned to prescriber’s practices, and adapt accordingly to address antibiotic misuse to curb antimicrobial resistance.

## 1. Introduction

During the onset of the COVID-19 pandemic in Singapore, public hospital admissions for community acquired acute respiratory tract infection (CA-ARI) had increased as patients were admitted for observation while awaiting confirmatory COVID-19 tests [1,2]. At Singapore General Hospital (a 1785-bedded acute tertiary care hospital), many hospital wards were converted to respiratory surveillance wards (RSWs) to segregate patients with respiratory symptoms until COVID-19 was excluded; confirmed cases of COVID-19 infection were transferred to dedicated COVID-19 isolation facility within the same campus [3]. Patients admitted for CA-ARI to the RSWs were often prescribed antibiotics that are commonly used to treat community-acquired pneumonia (CAP), namely amoxicillin-clavulanate, ceftriaxone, or oral levofloxacin. Increased consumption of respiratory antibiotics was observed [4], raising concerns for antibiotic misuse. Based on in-house observations, most of the patients with CA-ARI had no significant co-morbidities, and their workup often does not suggest a bacterial infection. In spite of this, antibiotics were frequently continued. Biomarkers such as procalcitonin may be used to distinguish between bacterial and viral infections [5,6,7,8], but treating teams may persist with antibiotic therapy despite reassuring procalcitonin levels. This was also compounded by the high patient load, rapidly changing guidelines and that the investigation results could not be acted upon in real time.

The main stewardship strategy by the Singapore General Hospital antimicrobial stewardship unit (ASU) has always been prospective audit and feedback (PAF). Pre-pandemic, piperacillin-tazobactam, carbapenems and intravenous fluoroquinolones, but not CAP antibiotics, were audited. With the pandemic, the increase in consumption of CAP antibiotics based on in-house antibiotic surveillance data prompted the ASU to extend PAF to CAP antibiotics prescribed in the RSWs, namely ceftriaxone, amoxicillin-clavulanate and oral fluoroquinolones. The macrolides, including azithromycin, were not included, as they are not mandatory to be added to ceftriaxone or amoxicillin-clavulanate when treating for a respiratory source of infection as per our hospital guidelines. As it was not feasible to audit all patients admitted to the RSWs who were prescribed CAP antibiotics, ASU identified the “high-yield” intervention population by utilizing procalcitonin (<0.5 µg/L) to identify patients with low likelihood of bacterial infections. Procalcitonin is a biomarker used to distinguish between bacterial and viral infection, and has been shown to reduce antibiotic consumption in patients with ARIs [6,7,8]. Although there is no well-established procalcitonin cut-offs to distinguish bacterial infection from viral infections [8,9], including COVID-19 [10,11,12], doctors in Singapore General Hospital typically use procalcitonin to guide antibiotic duration and ASU often apply a threshold of <0.5 µg/L for antibiotic discontinuation [13]. Patients without available procalcitonin levels were excluded from this review, as physicians do not order it routinely if there is strong clinical suspicion of a bacterial infection, e.g., compatible infective syndrome with leucocytosis, or neutrophilia. In these cases, antibiotics are usually warranted. Patients with elevated procalcitonin were also excluded, due to the possibility of a bacterial infection, which requires antibiotic treatment.

We hypothesized that expanding PAF to CAP antibiotics in this “high yield” intervention population would be strategic and relevant given the pandemic situation, potentially reducing unnecessary antibiotic prescriptions, with good safety outcomes. The primary objective of our study was to determine if PAF increases the proportion of patients with antibiotics discontinued within 4 days of initiation. Secondary objectives include evaluating the impact of PAF on duration of antibiotic therapy and the safety outcomes of patients in whom ASU intervened to discontinue antibiotics.

## 2. Results

There were 705 patients in the “pre-implementation” period, and 1768 patients in the “post-implementation” period who were admitted to RSWs and received CAP antibiotics. (see Figure 1). Patients who did not have a procalcitonin performed or had procalcitonin ≥0.5 µg/L were excluded from analysis (n = 335 and n = 891 in the “pre-implementation” and “post-implementation” period respectively). Amongst patients with procalcitonin <0.5 µg/L, 231 and 496 patients pre-and post-implementation, respectively, were excluded based on the study exclusion criteria. A total of 520 patients were included in the final analysis: 139 patients in the “pre-implementation” and 381 patients in the “post-implementation” period. PAF was conducted in 244/381 (64.0%) in the “post-implementation” period; PAF was not conducted in 137/381 (36.0%) patients because patients were moved out of the RSWs and lost to ASU purview. In addition, because ASU operates only on weekdays, there were missed opportunities for the audit and discontinuation of inappropriate CAP antibiotics over the weekend.

### 2.1. Patient Demographics and Baseline Characteristics

The baseline characteristics of the patients included for analysis are described in Table 1. The patients were largely similar in demographics, except for a higher Charlson’s co-morbidity score in the “post-implementation” period. Based on the distribution of biomarkers (see Table 1), the likelihood of bacterial infection was low in a significant proportion of patients.

In the “pre-implementation” and “post-implementation” periods, 56/139 (40.3%) and 137/381 (36.0%) of the patients had undetectable procalcitonin (<0.06 µg/L), respectively. Nearly two-thirds of the patients in each group had C-reactive protein levels <20 mg/mL, where bacterial ARI is unlikely. Less than half of the patients in each period presented with leucocytosis and less than a quarter had neutrophilia.

For the majority of the patients, microbiological investigations were unyielding (see Table 1). None of the patients in the “pre-implementation” and “post-implementation” period had positive cultures, except for one patient in the “post-implementation” period who had Pseudomonas aeroginosa isolated from the respiratory culture despite normal procalcitonin levels. A minority of patients had laboratory-confirmed respiratory viral infection (17/139 (12.2%) and 5/381 (1.3%), *p* < 0.001 in the “pre-implementation” and “post-implementation” periods respectively); COVID-19 infection was detected in 9 and 1 patients in the “pre-implementation” and “post-implementation” periods, respectively.

### 2.2. Primary and Secondary Outcomes

The primary and secondary outcomes are detailed in Table 2. There was a significantly higher proportion of patients who had antibiotics discontinued within 4 days, and a shorter overall duration of antibiotic therapy in the “post-implementation” period.

In a subgroup analysis of patients who received only intravenous antibiotics, there was a significantly shorter duration of therapy and shorter length of hospitalization in the “post-implementation” period. Notably, there was also a significant reduction in the proportion of patients whose intravenous antibiotics were switched to oral antibiotics.

### 2.3. ASU Interventions during “Post-Implementation” Period

In the “post-implementation” period, ASU intervened to discontinue antibiotics in 55/137 (40.1%) of the PAF patients. Amongst the 55 patients, 53 patients had respiratory symptoms or fever attributed to non-bacterial causes (e.g., viral infections, fluid overload, underlying structural lung disease and malignancy). The remaining 2 patients were appropriately started on antibiotics for bacterial pneumonia but had improved quickly by day 5 and ASU intervened for an earlier discontinuation of antibiotics.

The proportion of interventions accepted by the physicians was 46/55 (83.6%), and antibiotics were stopped within 48 h of intervention placement. Nine interventions to discontinue antibiotics were rejected; the treating physicians were concerned that all of these patients would deteriorate if antibiotics were discontinued.

### 2.4. Safety Outcomes

None of the patients who had ASU interventions placed had antibiotics reinitiated within 48 h for a respiratory tract infection, neither were they re-admitted for a respiratory tract infection within 30 days, and none died from a respiratory tract infection within 30 days of acceptance of ASU intervention.

Although there were 3 patients re-initiated on antibiotics within 48 h of ASU intervention acceptance, none were for respiratory tract infections (one received antibiotics for *Clostridioides difficile* diarrhoea, another received antibiotics for surgical prophylaxis, and the last patient had received antibiotics inappropriately for fluid overload).

## 3. Discussion

During the early phases of the COVID-19 pandemic, it has been observed that antibiotics were prescribed more indiscriminately for respiratory tract infections globally, both in the acute hospitals as well as in the community setting. In Spain, the use of ceftriaxone and azithromycin peaked in March and April 2020, similar to the observations in Singapore General Hospital [4,14,15]. As highlighted by a group of general practitioners in a focused group discussion conducted in the United Kingdom, there was a lower threshold for initiating antibiotics for respiratory tract infections in a pandemic, especially when the cause was unclear [16]. Our in-house surveillance data also revealed an increase in antibiotic consumption, in particular for antibiotics that were not usually audited by the ASU. Although a formal survey of the prescribers in our institution was not conducted, we suspect our prescribers had similar perceptions toward antibiotic use. The initial unfamiliarity with COVID-19 infection and concerns about bacterial co-infection resulted in prescribers “over-treating” their patients with antibiotics for a viral ARI [17]. Coupled with competing demands of high clinical workload and constantly evolving clinical workflows, prescribers had the tendency to administer antibiotics to many patients with CA-ARI instead of taking time to evaluate if their patients truly needed them.

It is well known that indiscriminate use of antibiotics is a known driver of antimicrobial resistance, and it would fuel a “dual COVID-19 and antimicrobial resistance pandemic” if left unchecked [18,19,20,21,22,23,24,25]. Hence, various centres have adapted their stewardship strategies for the pandemic and tailored their activities accordingly. For example, in Israel, Henig et al. described how they shifted the stewardship focus to the COVID-19 departments. In that setting, an intense stewardship program resulted in reduced antibiotic consumption through implementing new local guidelines and providing daily input from infectious disease consultants [26].

With Singapore General Hospital’s senior management support in maintaining stewardship activities throughout this pandemic and upon noticing the rising consumption of antibiotics, we promptly expanded the list of antibiotics for PAF to the CAP antibiotics, targeting patients admitted to the RSWs who were not under the purview of infectious diseases [4]. The expansion of PAF was also an opportune moment to pilot “disease-based” or “syndromic-based” stewardship which has already been described in other institutions [27]. Going forward, we plan to enhance our computer decision support system that incorporates patients’ clinical and laboratory data to automatically filter out “high-yield” cases for stewardship review and intervention. Additionally, with better machine learning capabilities in future, prompts could be designed to alert prescribers to reconsider the use of antibiotics for patients with low likelihood of bacterial infection [28].

Our stewardship strategy relied primarily on the procalcitonin threshold of 0.5 μg/L to identify cases for ASU audit in the RSWs. Although the optimal procalcitonin threshold limits are not well defined [5], this threshold appears reasonable in this small study, with none of our patients requiring antibiotics reinstated 48 h after acceptance of ASU intervention, neither were there 30-day readmissions nor 30-day mortality for respiratory infections. However, it is important to note that beyond looking at the procalcitonin levels, patients were evaluated holistically in terms of their clinical status and other biochemical parameters, e.g., C-reactive protein, leukocyte and differential count. One limitation of identifying patients for PAF using procalcitonin levels was that procalcitonin was not a mandatory test for CA-ARI management. Approximately 20% of the patients admitted to the RSWs and who were prescribed CAP antibiotics did not have procalcitonin tested, and these patients could have contributed to missed stewardship opportunities. Nonetheless, the approach of using procalcitonin to identify potential patients whose antibiotics were inappropriate was indeed “high-yield” and appropriate; 55/244 (23%) patients had an intervention by ASU to discontinue antibiotics, as compared to reviewing more than a thousand patients regardless of procalcitonin levels and intervening in less than 5% of the population.

The ASU intervention acceptance rate by the treating physicians to discontinue antibiotics was high (44/55, 83.6%). Nine recommendations were rejected because the primary physicians were concerned that their patients would deteriorate without antibiotics. This anxiety experienced by treating physicians has also been published in previous systemic reviews, where behavioural and belief systems influenced antibiotic prescriptions [29,30,31], Hence, ASUs have to incorporate behavioural change techniques in stewardship interventions to improve antibiotic prescribing behaviour. On hindsight, instead of placing interventions on the electronic medical records or simply texting the physicians, we could have reached out to the treating physicians (for 9 cases where ASU interventions were rejected) in the form of a face-to-face meeting, or phone conversation. However, this was not entirely feasible during the pandemic when there was strict segregation of staff and safe distancing measures in place. In addition, workload was heavy, and opportunities for direct communication either verbally or in person were missed. Other passive stewardship activities, such as developing and circulating educational modules for the management of ARIs, may have been useful but could not be conducted due to manpower and time constraints.

It appears that antibiotic prescribing patterns changed as the pandemic evolved. Four weeks after expanding PAF to CAP antibiotics in the RSWs, there was a decrease in eligible CA-ARI cases for ASU audit (Figure 2), and a significant increase in the appropriateness of antibiotic prescriptions in the RSWs as demonstrated by the reduction in ASU interventions to discontinued antibiotics over time. Antibiotics were proactively discontinued by the treating physicians even before ASU review, even in patients with a higher Charlson’s co-morbidity score in the “post-implementation” period. The findings of this study have demonstrated that the presence of ASU is important and has a positive direct and indirect impact on the appropriateness of antibiotic prescriptions. ASU interventions could also be perceived as a form of physician education. Undoubtedly, prescribers could have gained more confidence in managing patients with CA-ARI as the pandemic evolved and through other educational platforms, including regular townhalls conducted by the hospital leadership team in collaboration with the infectious diseases team. It is possible that the compulsion to start antibiotics out of fear was reduced as the physicians were empowered [16].

In the majority of the patients with CA-ARI, a specific cause could not be identified. For all the cases, only COVID-19 testing was mandatory, while the other microbiological investigations, including respiratory virus panel polymerase chain reaction, were performed only upon request by the treating physicians and as clinically indicated. Laboratory-confirmed infections were positive only in a minority of the patients, and were mostly due to respiratory viruses. Notably, there were even fewer non-COVID-19 respiratory viruses detected in the “post-implementation” period, which corroborates with the trends observed nation-wide during the COVID-19 pandemic in Singapore [32]. Possible reasons include the following: (1) physicians were discouraged to order respiratory multiplex virus polymerase chain reaction tests to conserve resources for COVID-19 testing, and (2) reduced transmission of other respiratory virus CA-ARIs due to lockdown measures, universal mask mandate, and international travel restrictions. Additionally, only a handful of patients were tested positive for COVID-19 infections in our study as patients who were tested positive for COVID-19 at the emergency department were already directly transferred to the COVID-19 isolation wards. The COVID-19 cases that the RSWs subsequently identified were probably missed during the initial screening.

## 4. Materials and Methods

### 4.1. Study Population

This was a retrospective single-center pre-post quality improvement study conducted in Singapore General Hospital to evaluate the impact of conducting PAF on CAP antibiotics in patients with CA-ARI during the initial wave of the COVID-19 pandemic. CA-ARI was defined as a syndrome consisting of fever, elevated inflammatory markers, or respiratory symptoms, such as rhinorrhoea, sore throat, dyspnoea and cough, or more than one of the above.

Patients with CA-ARI admitted to RSWs within 48 h of admission, prescribed CAP antibiotics for CA-ARI and had normal procalcitonin (<0.5 µg/L) were identified electronically and screened for study eligibility. The following patients were excluded from further analysis: those who had non-respiratory infections in addition to CA-ARI; admitted to the intensive care unit; died within 48 h of admission; diagnosed with pulmonary tuberculosis; had radiological findings suggestive of complicated pulmonary infections (e.g., lung abscesses); or required escalation of therapy to anti-pseudomonal antibiotics. Patient demographics, laboratory, microbiological investigations and clinical progress were obtained from the electronic records.

### 4.2. ASU’s Intervention

The “pre-implementation” period was defined as the 4-week period between 22 March 2020 and 18 April 2020, and the “post-implementation” period was conducted for 12-weeks between 21 April 2020 and 13 July 2020. During the “post-implementation” period, ASU’s PAF was expanded to ceftriaxone, amoxicillin-clavulanate, and oral levofloxacin. PAF was conducted every weekday by the ASU pharmacists on the second or third day of CAP antibiotic initiation in the “high-yield” intervention patient population described earlier. This patient list was electronically generated from our in-house electronic health information system. ASU pharmacists evaluated the antibiotics for appropriateness based on indication, choice, route of administration, and duration of therapy through holistic chart review. For patients with inappropriate antibiotic prescriptions, the ASU pharmacists either intervened to provide therapeutic recommendations to the primary physicians directly, or discussed complex cases with diagnostic conundrum with rostered ASU infectious disease physician prior to intervening. All interventions were followed up for acceptance within 48 h.

### 4.3. Study Outcomes

The primary outcome was the proportion of patients who had antibiotics discontinued within 4 days of initiation. The secondary outcomes were overall duration of antibiotic therapy and proportion of patients with intravenous-to-oral switch in antibiotics. A subgroup analysis was also performed on patients who received only intravenous antibiotics; the duration of antibiotics and length of hospitalization were compared between groups. We also compared the following safety indicators between patients whose interventions were accepted or rejected: (a) the proportion of antibiotic re-initiation for respiratory tract infections within 48 h of intervention acceptance, (b) 30-day re-admission rates for respiratory tract infections, and (c) mortality rates from a respiratory tract infection within 30 days of ASU intervention.

Categorical variables were compared using either chi-squared test or Fischer exact (two-tailed) test. Continuous variables were all not normally distributed and expressed as median (inter-quartile range). The 2 groups were analysed using Mann–Whitney U test. All statistical analysis was conducted using IBM SPSS Statistics for Windows, Version 20.0 (IBM Corp., Armonk, NY, USA).

## 5. Conclusions

The COVID-19 has challenged us to rethink our stewardship approach. While PAF remains our core strategy, antibiotics targeted for audit may require modifications depending on changes in prescribers’ habits. Our in-house antibiotic consumption surveillance system allowed the team to response to changes in antibiotic use during the pandemic and we responded promptly. The availability of information technology streamlined our workflow processes and improved the efficiency of the ASU. Thus, allowing the expansion of our audit despite resource limitations during the pandemic. This study demonstrates the importance of ASU during the pandemic, and the need for ASU to remain attuned to prescriber’s practices, identify situations where there could be potential for antibiotic misuse, and adapt existing approaches to curb antimicrobial resistance.

## Figures and Tables

**Figure 1 antibiotics-11-00846-f001:**
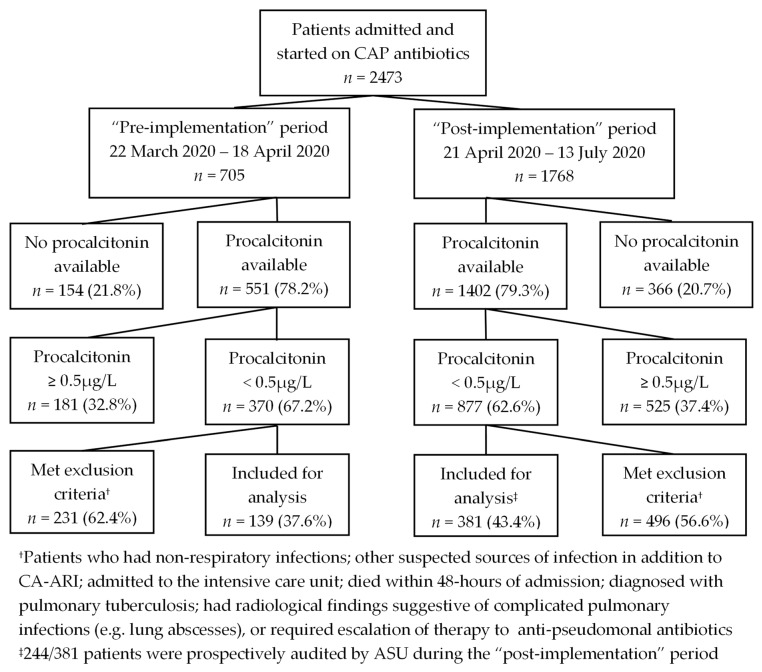
Patients with community-acquired respiratory infections admitted to the respiratory surveillance wards were screened for suitability for stewardship audits. Abbreviations: CAP, community-acquired pneumonia; CA-ARI, community-acquired acute respiratory infection.

**Figure 2 antibiotics-11-00846-f002:**
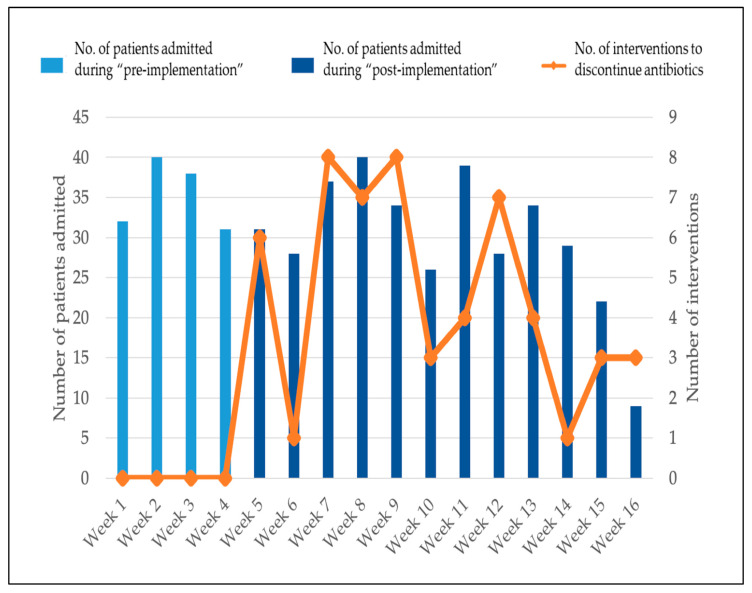
Weekly number of patients admitted to the RSWs who fulfilled study criteria (left axis) and the weekly number of ASU interventions made to discontinue antibiotics (right axis). Audit of CAP antibiotics were not performed in the “pre-implementation” period (Weeks 1 to 4) and there were no stewardship interventions placed during that time.

**Table 1 antibiotics-11-00846-t001:** Baseline demographics and characteristics of patients admitted for community acquired respiratory infection during the initial wave of the COVID-19 pandemic in Singapore General Hospital.

	Pre-Implementation Period (22 March 2020–18 April 2020) *n* = 139	Post-Implementation Period (21 April 2020–13 July 2020) *n* = 381	*p*-Value
**Patient demographics, median [IQR] or *n*(%)**			
Age in years	68 (56–81)	71 (57–82)	0.270
Male	67 (48.2)	216 (56.7)	0.085
Charlson’s comorbidity index	4 (2–6)	5 (2–7)	<0.001
Congestive heart failure	15 (10.8)	50 (13.1)	0.477
Chronic kidney disease, stages 4–5 or receiving dialysis	8 (5.8)	41 (10.8)	0.084
Lung malignancy	14 (10.1)	33 (8.7)	0.620
Underlying structural lung disease (COPD/bronchiectasis)	19 (13.7)	51 (13.4)	0.933
**Biochemical parameters ^a^, median [IQR] or *n* (%)**			
Procalcitonin in µg/L	0.07 (0.06–0.13)	0.07 (0.06–0.15)	0.715
Patients with undetectable procalcitonin (<0.06 µg/L)	56 (40.3)	137 (36.0)	0.366
C-reactive protein in mg/L	12.5 (2.30–51.00)	11.2 (3.05–44.95)	0.781
Patients with C-reactive protein <20 mg/L	73/117 (62.4)	212/346 (61.3)	0.829
White blood cells × 10^9^/L	8.75 (6.26–11.52)	8.82 (7.04–11.57)	0.364
Patients with white blood cells <10 × 10^9^/L	79/136 (58.1)	228/369 (61.8)	0.450
Neutrophil differential in %	71.9 (62.08–79.90)	72.7 (63.15–79.85)	0.452
Patients with neutrophils differential <80%	104/136 (76.5)	280/369 (75.9)	0.890
**Microbiological investigations, *n* (%)**			
Laboratory confirmed respiratory viral infection ^b^ using respiratory panel RT-PCR assays ^c^	17 (12.2)	5 (1.3)	<0.001
SARS-CoV-2	9	1	
Influenza A	1	1	
Rhinovirus	3	0	
Metapneumovirus	2	0	
Adenovirus	1	2	
Human coronavirus OC43	1	0	
Respiratory syncytial virus	0	1	
Positive respiratory cultures	0 (0.0)	1 ^d^ (0.3)	1.000

Abbreviations: IQR inter-quartile range; COPD chronic obstructive pulmonary disease; RT-PCR respiratory tract polymerase chain reaction. Footnote: ^a^ Based on laboratory investigation results available within 2 days of admission date. C-reactive protein was available in 117 and 346 patients in the “pre-implementation” and “post-implementation” period respectively. White blood cell was available in 136 and 369 patients in the “pre-implementation” and “post-implementation” period respectively; ^b^ Respiratory virus multiplex PCR (qualitative) was performed using AnyplexTM II RV16 Version 1.1 (Seegene, Seoul, Korea); ^c^ SARS-CoV-2 PCR was performed using Xpert Version 1.3 (Cepheid, Sunnyvale, USA) and COBAS 6800 Version 4.7b (Roche Diagnostics, Mannheim, Germany) platforms; ^d^ One patient, who had a renal transplant, received three days of oral amoxicillin-clavulanate prior to admission and sputum culture isolated *Pseudomonas aeroginosa*.

**Table 2 antibiotics-11-00846-t002:** Primary and secondary outcomes of the study.

	Pre-Implementation Period (22 March 2020–18 April 2020) *n* = 139	Post-Implementation Period (21 April 2020–13 July 2020) *n* = 381	*p*-Value
**Primary outcome**			
Patients with antibiotics discontinued within 4-days, *n* (%)	23 (16.5)	133 (34.9)	<0.001
**Secondary outcomes**			
Overall duration of antibiotic therapy in days, median [IQR]	7 (6–8)	6 (3–8)	<0.001
Patients with IV-to-PO switch of antibiotics, *n* (%)	63 (45.3)	131 (34.4)	<0.05
Patients receiving IV antibiotics only, *n* (%)	17 (12.2)	75 (19.7)	<0.05
Corresponding duration of therapy in days, median [IQR]	3 (2–8)	2 (1–5)	<0.05
Corresponding length of hospitalization in days, median [IQR]	10 (3.5–16)	5 (3–10)	0.058

Abbreviations: IQR inter-quartile range; IV intravenous; PO per orally.

## Data Availability

The data presented in this study are available on request from the corresponding author (S.J.C.) upon reasonable request.

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
