# Peer review of "Stemming the Rise of Antibiotic Use for Community-Acquired Acute Respiratory Infections during COVID-19 Pandemic"

_antibiotics, 2022, doi:10.3390/antibiotics11070846_

Round 1

Reviewer 1 Report

Dear Editor,

The article entitled “Stemming the rise of antibiotic use for community-acquired acute respiratory infections during COVID-19 pandemic” (manuscript: antibiotics-1770271) concerns the use or misuse of antibiotics for the treatment of community acquired respiratory tract infection (CA-ARI) in patients admitted for suspected or confirmed COVID-19. It was done in Singapore (Singapore General Hospital, SGH) and underlines the importance of antimicrobial stewardship unit to remain attuned to prescriber’s practices, identify situations where there could be potential for antibiotic misuse and adapt existing approaches to curb antimicrobial resistance.

The paper is well written and detailed: it needs minor revisions.

I suggest to explain that procalcitonin is used to distinguish viral from bacterial pneumonia in the abstract and also mention a recent review about this topic: Kamat, I. S., Ramachandran, V., Eswaran, H., Guffey, D., & Musher, D. M. (2020). Procalcitonin to distinguish viral from bacterial pneumonia: a systematic review and meta-analysis. Clinical Infectious Diseases, 70(3), 538-542.

Moreover, there are some recent interesting paper that may be mentioned, regarding COVID-19 pandemic (Iacopetta, D., et al. COVID-19 at a glance: an up-to-date overview on variants, drug design and therapies Viruses, 2022, 14(3), 573) and antimicrobial resistance (Catalano, A., et al. Multidrug Resistance (MDR): A Widespread Phenomenon in Pharmacological Therapies. Molecules, 2022, 27(3), 616).

In the abstract I would delete numbers 139 and 381. I suggest to change “16.5% (23/139) vs. 34.9% (133/381)” into “16.5% vs. 34.9%” and “45.3% (63/139) vs. 34.4% (131/381)” into “45.3% vs. 34.4%”.

Some minor corrections:

Use “high-yield intervention” or “high-yield” intervention (througout the whole text)

I suggest to add Abbreviations at the end of the text

Line 139 Forty-six (83.6%) interventions: add the number and not only percentage or delete percentage.

Line 208-209: use percentage or numbers (83.6%…. Nine)

C. difficile”: use the correct name

Author Response

Reviewer 1

Feedback 1. Explain procalcitonin is used to distinguish viral from bacterial pneumonia in abstract, and also mention a recent review about this topic (Kamat 2020)

Response: Thank you for your suggestion. We have included a brief explanation in the abstract (in lines 14-16) and cited Kamat 2020 in the introduction (Line 45), as well as in the discussion (Line 205).

Feedback 2. Some recent interesting paper that may be mentioned regarding COVID-19: Iacopetta 2022 and Catalano 2022.

Response: We have included both references in the discussion [line 184]

Feedback 3. Remove numbers 139 and 381 in the abstract

Response: We have presented the results in percentages only for the abstract.

Feedback 4. Consistency in “high-yield intervention” or “high-yield” intervention, throughout whole text

Response: Thank you for pointing this out. We have adjusted to use “high-yield” intervention consistently throughout the whole text.

Feedback 5. Add abbreviations at the end of the text

Response: Thank you for your suggestion, we have included a section for the abbreviations used at the end of the manuscript.

Feedback 6. Line 139 Forty-six (83.6%) interventions: add the number and not only percentage, or delete percentage

Response: We have rephrased the sentence to include the total number of interventions i.e. 46/55, 83.6%

Feedback 7. Line 208-209: use percentage or numbers (83.6% … Nine)

Response: We have rephrased the sentence to present the data as 46/55, 83.6%, to present the data in the same format through the text. Similar to Feedback 6.

Suggestion 8. Use the correct name for C difficile

Response: Thank you for pointing this out. We have changed it to Clostridioides difficile.

Reviewer 2 Report

v

Author Response

Reviewer 2:

Feedback 1. The number of patients included in the “post-implementation” period. In your results section, you wrote: “PAF was conducted in 137/38 (3.0%) in the “post-implementation” period.” According to Table 1, we can hypothesize that you include only 137 patients with undetectable procalcitonin while the inclusion criterion was patients with normal procalcitonin. Why only 137 patients received PAF among 381 included?

Response: Thank you for pointing this out.

We would take the opportunity to clarify. There were 381 patients available for analysis (meaning, they qualified for ASU audits, based on the inclusion criteria were listed in the methods section and as highlighted by Figure 1).

By criteria, 381 patients qualified for ASU audits, however, we were only able to perform PAF on 244 patients. This is not uncommon in our practice. Not all antibiotic prescriptions were eventually audited and/or intervened upon by the ASU for reasons we will elaborate in the paragraph below.

Specifically for the stipulated time period of the study (and this is pertaining to the 381 cases), the cases admitted to the respiratory surveillance wards (RSWs) were dynamic; some of these patients were moved out of RSWs very quickly. This meant that these patients were lost to ASU purview, and PAF with interventions may not be performed. In other instances, antibiotics may be discontinued by the primary physicians prior to ASU interventions, OR, a course of antibiotics may have been initiated or continued over the weekend (and ASU does not perform prospective audit and feedback during this time); by the next working day, the course of antibiotics may have been completed. These represent lost opportunities for intervention. Unfortunately, this is one of the limitations of our system. This was discussed in Lines 89-93.

Fortuitously, 137/381 patients also had a procalcitonin level of <0.06mg/L, but may or may not have had ASU audit.

Feedback 2: CIRB approval is dated in 2010 before the COVID-19 pandemic

Response: Thank you for raising this concern. Singapore General Hospital’s antimicrobial stewardship unit had obtained SingHealth Centralized Institutional Review Board’s approval to conduct all ASU related research in 2010, including research pertaining to prospective audit feedback for quality improvement purposes. We have been exempted from the renewal of IRB for all stewardship related work subsequently.

We have amended the manuscript to improve clarity.

Feedback 3: Regarding abbreviations used in the text, a recommendations was made to use less abbreviations for better readability.

Response: Thank you for your recommendations. We have removed the abbreviations “SGH”, “ID”, “AMR” and “IT” and replaced them with the actual words.

Feedback 4: Why were 19-20 April 2020 not included in the study period?

Response: Due to the evolving COVID-19 pandemic, ward conversions were dynamic. There were deliberations and discussions on bed situation by the hospital management, and there was uncertainty on conversion of regular ward to RSWs, on 19-20 April 2020. Hence we decided to include patients from 21 April 2021 when RSW ward locations were better defined and more stable.

Feedback 5: Why patients with bacteraemia were included, even though we specified that patients were excluded if they had more than one site of infection?

Response: We would like to thank the reviewer for pointing this out. The 2 cases of bacteraemia in the post intervention period listed in the original manuscript had presented with fever but had paucity of respiratory symptoms; regardless they fulfilled the admission criteria for CA-ARI and were sent to the RSWs. It was only later that the bacteremia was picked up. For these 2 cases, the E. coli and S. maltophilia bacteremia upon adjudication could not be attributed to a respiratory tract infection with confidence. As such, we have removed the information from Table 1, and updated the text in the manuscript accordingly.

Feedback 6: Line 101-102, 130/381 (34.1%) of the patients had undetectable procalcitonin (<0.06 µg/L), but was not in accordance to Table 1.

Response: Thank you for pointing out this typographical error: It should be 137/381 (36.0%) instead of 130/381 (34.1%). We have amended this in the manuscript.

Feedback 7: Line 104: to remove references [13, 14] under results

Response: Thank you for your suggestion, we have removed the aforementioned references.

Feedback 8: Did we use different computer systems in this study?

Response: Yes, we used different information technology systems to help us in different ways.

Computer decision support system is for prescribers. It is embedded in our electronic prescribing system where prescribers have to select the antibiotic options available to reflect the type and severity of infection of the patients whom they are ordering antibiotics for.

In contrast, the in-house electronic health information system (e-HINTS) is for analytics. This system is a database of patients admitted in our hospital, and includes data such as admission / discharge dates, ward locations, laboratory tests/results etc. The ASU team uses this system to monitor antibiotic use, and review trends in regard to antibiotic prescribing habits.

The in-house antibiotic consumption surveillance system is a dashboard showing the proportion of patients on antibiotics at any one time, and may be filtered based on department or ward locations.

Feedback 9: For Table 1, specify the exact location & improving readability of the table

Response: Thank you for your suggestion. We have specified the exact location “in Singapore General Hospital” in the title and adjusted Table 1 to improve readability.

Feedback 10: Table 2, patients with antibiotics discontinued within 4-days in the “post-implementation” period among 137 patients with PAF. Explain reason for the four patients with continued antibiotic prescriptions.

Response: Thank you for your question. As mentioned in response to feedback 1, a total of 381 patients were eligible for ASU audits in the “post-implementation” period. However, 244/381 patients were prospectively audited by ASU due to limitations of our system.

Amongst the 381 patients, 133 patients had antibiotics discontinued within 4-days; they may or may not have been prospectively audited by ASU.

Feedback 11: Regarding the safety outcomes subsection on length of hospitalization as secondary outcome? You presented the length of hospitalization outcome only for patients requiring IV antibiotics (Table 2), this supplementary outcome could be interesting among safety outcomes for all patients in pre/post-implementation periods no?

Response: Based on our inhouse surveillance data over the last decade, the length of hospitalization is dependent on the use of intravenous antibiotics. Specifically for CA-ARI during the initial wave of the pandemic, relatively well patients were admitted for surveillance and monitoring purposes. A significant proportion of these patients were placed on oral antibiotics, and could be discharged for home isolation where appropriate(this data was not presented in the text). Because many patients in the pre/post-implementation periods had mild infections and most of them were on oral antibiotics, we felt that this would not be a useful or interesting efficacy / safety outcome to track given the presence of non-antibiotic-related confounders, and it would not affect the length of hospitalization. In contrast, the use of intravenous antibiotics invariably affects length of stay and is a marker of more severe infections. This, in our opinion, would be a more appropriate efficacy / safety outcome indicator to report on ASU interventions. Thence, the decision to look at the length of hospitalization only in patients who received intravenous antibiotics.

Feedback 12: Regarding statistical analysis, if all the data had non-normal distribution and to revise the statistical analysis section accordingly

Response: Thank you for your suggestion. All the continuous data was not normally distributed. We have adjusted the statistical analysis section accordingly and removed the statistical tests for normally distributed data.

Feedback 13: Figure 2 was presented in the methods section but cited in the discussion section. Moreover, the two figures can be combined in a simple figure to improve readability and to show proportion of patients with ASU interventions placed to discontinue antibiotics

Response: Thank you for your suggestion! We have shifted Figure 2 to the discussion section, closer to where it was cited, and combined the two graphs. We have presented the number of intervention as a line chart, superimposed on the bar charts to provide greater clarity of the trends.

Round 2

Reviewer 2 Report

Thank you for the revision and for the explanations in your response letter.

The presentation of the number of intervention as a line chart, superimposed on the bar charts in the figure 2 was a great idea to improve the clarity of the trends.

Two minor revisions need to be considered:

1) Line 92: remove the sentence "We acknowledge that this is one of the limitations of the system." which is a discussion sentence, not a result sentence. The paragraph (line 89-92) you have added following my previous review is perfectly clear; this last sentence is not necessary.

2) Line 235: replace "(Figure 2a)" by "(Figure 2); and line 237 remove "(Figure 2b)".

Congratulations on your work.

Author Response

Dear Reviewer, 

Thank you for your comments and your recommendations. 

Two minor revisions need to be considered:

For the points raised, replies in italics. 

1) Line 92: remove the sentence "We acknowledge that this is one of the limitations of the system." which is a discussion sentence, not a result sentence. The paragraph (line 89-92) you have added following my previous review is perfectly clear; this last sentence is not necessary.

Thank you. We have removed lines 92. The changes does not appear on the revised manuscript due to track changes.

2) Line 235: replace "(Figure 2a)" by "(Figure 2); and line 237 remove "(Figure 2b)".

The amendments have been made. (Figure 2a) is now replaced by (Figure 2). We have also removed (Figure 2b).